

# Epibiotic *Vibrio* bacteria from crustaceans and macroalgae in a subantarctic ecosystem, and their future thermal suitability

Manuel Ochoa-Sánchez[1,2], Rosalinda Tapia-López[1,2], Ulises E. Rodriguez-Cruz[1], Eliana Paola Acuña Gomez[2], Luis E. Eguiarte[1] and Valeria Souza[1,2]

[1] Departamento de Ecología Evolutiva, Instituto de Ecología, Universidad Nacional Autónoma de México, CDMX, Mexico
[2] Centro de Estudios del Cuaternario de Fuego, Patagonia y Antártica (CEQUA), Punta Arenas, Chile

## ABSTRACT

Marine organisms harbor diverse microbial communities on their surface, yet studies exploring the epibiotic bacteria of marine hosts remain largely unexplored, particularly in subantarctic ecosystems. Here, we cultured and isolated bacteria from the surface of three marine hosts: the centolla (the southern king crab; *Lithodes santolla*), a squat lobster (*Grimothea gregaria*), and a brown macroalgae (*Macrocystis pyrifera*), from a subantarctic ecosystem, the Magellan Strait. Bacteria were inoculated in Petri dishes with Thiosulfate-Citrate-Bile Salts-Sucrose (TCBS) agar medium, and a fragment of the grown colonies was used to extract their DNA and sequence the whole 16S rRNA gene. We detected 14 different bacterial taxa, 11 from crustaceans, most of which were found only in the squat lobster. *Vibrio* spp. was detected in all marine hosts, but *V. tasmaniensis* was only detected in crustaceans. Phylogenetic comparisons revealed that epibiotic *Vibrio* formed a clade related to environmental *Vibrio* species, such as *V. tasmaniensis*, *V. echinoidereum*, and *V. atlanticus*. Given the ongoing climate change the world is experiencing, we explore the future sea surface temperatures that these bacteria might experience in the Magellan Strait. Oceanographic predictions indicate that the maximum sea surface temperatures will be 1 °C warmer in the future decades, and they could reach values above 14 °C in the last decades of the century. Our results increase the distribution and ecology of *Vibrio* bacteria and give insights about the temperatures that these microbes will face in future decades, which could have relevant consequences for aquaculture activities.

Corresponding authors
Manuel Ochoa-Sánchez,
manuel.ochoa@iecologia.unam.mx
Valeria Souza, souza@unam.mx,
souza.valeria2@gmail.com

## INTRODUCTION

During the last two decades, the average global sea surface temperature has increased by 1 °C (*Huang et al., 2024*). However, sea surface temperature has different behaviors in local contexts, particularly in the Southern Hemisphere, where temperature anomalies between 2–4 °C have lasted for several weeks (*Holland et al., 2021*). Increases in sea surface

temperature can threaten ocean ecosystem integrity, leading to a decline in habitat-forming species, like corals and their associated fish populations, as well as invertebrates and marine mammals (*Venegas, Acevedo & Treml, 2023*; *Ripple et al., 2025*). Also, increases in sea surface temperature are expected to be one of the main drivers of the increase in the geographic distribution of marine pathogens, resulting in higher rates of contaminated seafood and seafood-borne illnesses (*Rowley et al., 2024*). Moreover, climate change is thought to aggravate the frequency, severity, and geographic distribution of known diseases (*Ebi et al., 2017*; *Mora et al., 2022*), where coasts are particularly vulnerable ecosystems (*Romanello et al., 2023*). To inform about the pathogenic potential in local areas, it is imperative to have better knowledge of autochthonous biodiversity.

*Vibrio* bacteria are the most common cultivable microbes in marine and coastal waters (*Gomez-Gil et al., 2014*). Historically, *Vibrio* bacteria distribution has been restricted to tropical and temperate waters (*Vezzulli, Colwell & Pruzzo, 2013*). However, in the last decades, the incidence of *Vibrio* epidemic outbreaks has increased globally due to the increase in global sea surface temperature (*Baker-Austin et al., 2024*). Moreover, several studies have linked latitudinal increases in sea surface temperature with the latitudinal expansion of *Vibrio* bacteria in the Northern Hemisphere (*Baker-Austin et al., 2013*; *Vezzulli et al., 2015*; *Archer et al., 2023*). Also in the Southern Hemisphere, the latitudinal expansion of *Vibrio* bacteria has started, as shown by the recent reports of epidemic outbreaks of pathogenic *Vibrio* in southern latitudes (*Raszl et al., 2016*). Furthermore, *Vibrio* pathogenic outbreaks have been recorded in southern Chile, in Puerto Montt (41°28′) (*González-Escalona et al., 2005*) and in Reloncavi fjord (41°, 42′) (*Bacian et al., 2021*). However, we still miss information about native *Vibrio* bacteria in subantarctic ecosystems, such as the Magellan Strait.

The Magellan Strait, Chile, is an extensive geographical area (ca. 8,500 km of coastline), located in the southernmost region of South America. It is a V-shaped interoceanic channel, where Atlantic and Pacific Ocean waters converge (*Andrade, 1991*). The Magellan Strait is also seasonally influenced by freshwater input from precipitation and glacier melt in spring/summer (*Rivera et al., 2023*). Together, these characteristics create a complex oceanographic scenario that allows high seasonal productivity (*Aracena et al., 2011*; *Pantoja, Iriarte & Daneri, 2011*).

The epibiotic bacteria of marine hosts in the Magellan region, particularly from crustaceans, remain poorly studied (*Ochoa-Sánchez et al., 2023a*). This is an important gap since several hosts could harbor epibiotic *Vibrio* bacteria, such as the squat lobster (*Grimothea gregaria*, Fabricius, 1793), a key species in the trophic network of the region (*Haro et al., 2022*); centolla (*Lithodes santolla*, Molina, 1782), an abundant crustacean of economic importance (*Pollack, Berghöfer & Berghöfer, 2008*); and the brown macroalgae (*Macrocystis pyrifera*, L. C.Agardh, 1820), an ecosystem engineer that provides shelter for several species, including centolla (*Cárdenas et al., 2007*). Under the ongoing climate change, southern Chile will face warmer temperatures, which currently has caused consistent glacier retreat and thinning during the last 20 years (*Dussaillant, Berthier & Brun, 2019*). Furthermore, marine heatwaves (discrete warm water anomalies) have been increasing in frequency and duration during the last ten years (*Pujol et al., 2022*;

*González-Reyes et al., 2023*). Given the relationship between increased surface temperature and pathogen proliferation (*Rowley et al., 2024*), it is crucial to couple biological surveys of potential pathogens with their future thermal suitability.

In this study, we have three objectives: (1) Isolate and identify epibiotic *Vibrio* bacteria from the centolla (*Lithodes santolla*), the squat lobster (*Grimothea gregaria*), and the brown macroalgae (*Macrocystis pyrifera*); (2) determine if these *Vibrio* isolates are phylogenetically related to pathogenic *Vibrio* species; and (3) explore the future sea surface temperatures under the most plausible climate change scenarios that both hosts and their epibiotic bacteria will face in future decades. We found several bacterial taxa associated with the surface of each host. *Vibrio* spp. was detected in all marine hosts, but *V. tasmaniensis* was only detected in crustaceans. Also, *Shewanella* spp. was only detected in crustaceans, while *Photobacterium profundum* and *Psychromonas antarctica* were only detected in macroalgae. Phylogenetic comparisons revealed that epibiotic *Vibrio* formed a clade related to environmental *Vibrio* species, such as *V. tasmaniensis*, *V. echinoidereum*, and *V. atlanticus*. Average sea surface temperature is expected to be similar between contemporary times and climate change scenarios, while maximum sea surface temperature is expected to be significantly higher in future decades.

## MATERIAL AND METHODS

### Marine host sampling

Sampling was conducted on four individuals of centolla, four individuals of squat lobsters, and two individual macroalgae blades from Choiseul Bay in the Magellan Strait ($-53.59°$S, $-72.3°$W) during the austral summer (February 2022). Centollas were sampled by diving up to 12 m below sea level. In the case of macroalgae blades, the obverse of the floating blades was sampled. Squat lobsters were sampled from nocturnal swarms that arose near the ship. Between 10–15 squat lobsters were collected, but the sampled ones (four) were kept in separate containers with marine water to avoid fecal contamination since they defecate upon contact with each other. All samples were collected following approved bioethical guidelines of the Comité de Ética, Bioética y Bioseguridad, Universidad de Concepción, Chile (protocol number CEBB 1081 2021) and a sampling permit of the Subsecretaría de Pesca y Acuicultura, Chile (resolution E- 2021- 531).

### Epibiotic bacterial isolation and DNA extraction

We swabbed the body and legs of both crustacean species, as well as all the blade surfaces from each individual with one swab per individual (Puritan, Guilford, ME, USA), which later were used to conduct the basic striatum technique in prepared Petri dishes with Thiosulfate–citrate–bile salts–sucrose agar medium (BD TCBS$^{TM}$ DIFCO). Media swabbing was conducted near a laboratory gas burner inside the ship to prevent airborne contamination. TCBS agar medium is regarded as a selective medium for isolating *Vibrio* bacteria and related species since it has high concentrations of sodium thiosulfate and sodium citrate, inhibiting the growth of opportunistic Enterobacteria. Typically, grown colonies that belong to *Vibrio* are large and have yellow-green tones (*Pfeffer & Oliver, 2003*). Cultures were sealed with parafilm paper and placed inside the incubator, where

they were incubated at ca. 37 °C for four days to ensure enough colonies of a proper size to obtain genetic material. After incubation, colonies were kept in cold (5–6 °C) and dark conditions for two days, until arrival at CEQUA facilities, where they were stored in cold (4 °C) for 10 weeks until DNA extraction. Colonies were distributed across the plate and had a visible size. Swabs from all hosts retrieved yellow-orange colonies. After examination for the presence of yellow or green colonies, each colony was grown independently on the same media to isolate single colonies, and the plates were incubated overnight at 37 °C. Total genomic DNA was extracted from each single colony using the DNAeasy Blood and Tissue kit (QIAGEN)."

## Amplification of 16s rDNA and taxonomic annotation

Amplification of the 16S rDNA was conducted using the universal primers 27F (5′-AGA GTT TGA TCC TGG CTC AG-3′) and 1492R (5′-GGT TAC CTT GTT ACG ACT T-3′) (*Lane, 1991*). An amplification reaction was performed with 50 ng of genomic DNA at a final PCR reaction of 25ul (Go Taq Flexi DNA Polymerase; PROMEGA). The reaction mixture was amplified in a Thermal Cycler C100 Touch (Bio-Rad, Hercules, CA, USA) with the following conditions: denaturation at 94 °C for 5 min; followed by 30 cycles: 94 °C for 1 min for denaturation, 50 °C for 30 s for annealing, and extension at 72 °C for 1 min; and final extension at 72 °C for 5 min. PCR products obtained were ca. 1,500 bp, and their quantity and quality were first evaluated by electrophoresis in 1.0% agarose, and then quantified by Nanodrop. The PCR products were Sanger sequenced in both directions by Macrogen in South Korea.

The program 4 Peaks version 1.8 was used to view the quality of DNA sequences. Sequences were then reviewed, trimmed, assembled, and edited using Geneious Prime version 2020.5. Determination of the taxonomic assignment of each bacterial isolate was done using a BLAST search at NCBI (https://blast.ncbi.nlm.nih.gov/Blast.cgi). We reported the best-scoring reference sequence similarity cut-off of 98–100%. The 15 sequences obtained were then submitted to GenBank with the following accession numbers PP266351 to PP266365.

## Phylogenetic comparison of *Vibrio* isolated from Magallanes

Due to the large number of fully sequenced genomes of *Vibrio* spp. available in the NCBI database (746 as of November 20, 2024), 72 genomes classified as fully sequenced reference genomes were downloaded from the NCBI database (https://www.ncbi.nlm.nih.gov/datasets/genome/?taxon=662&reference_only=true&assembly_level=3:3).

For the phylogenetic comparison, we obtained the 16S rRNA gene from the complete genomes by using the software barrnap (v.0.9) (*Seemann, 2013*). Subsequently, the 16S rRNA genes from the 72 genomes, including the six isolates from Magallanes, were aligned using MAFFT (v.7.490) (*Katoh & Standley, 2013*). The phylogenetic tree was inferred using maximum likelihood as implemented in IQ-TREE (*Minh et al., 2020*). ModelFinder was used to select the sequence evolution model which was TIM3e+I+G4 (*Kalyaanamoorthy et al., 2017*). To estimate branch support, 1,000 ultrafast bootstrap replicates were applied to estimate branch support (*Hoang et al., 2018*). The tree was visualized using iTOL (*Letunic & Bork, 2021*).

### Sea surface temperature projections under climate change scenarios

Average and maximum sea surface temperature (SST max) were extracted from Bio-oracle database version 3, using its associated R package, *smdpredictors* (*Assis et al., 2024*; *Bosch & Fernandez, 2021*). We used average temperature as a proxy of the most common sea surface temperature in the Magellan Strait, but also maximum sea surface temperature as a proxy of the upper temperatures that the Strait could experience in the future. We considered the following scenarios: contemporary (average temperatures from 2000–2020) and the two most plausible climate change scenarios according to *Burgess et al. (2023)*; Shared Socioeconomic pathway (SSP) 2.6 and 4.5, at two temporal resolutions 2020–2050 and 2050–2100.

The geographic range was delimited to the Magellan Strait (−52.2, −68; −54.5, −73), where the samples were taken. Data management and posterior analysis were done in R version 4.1 (*R Core Team, 2021*). We projected 30 random points across the Magellan Strait (−52.2, −68; −54.5, −73) to extract average and maximum sea surface temperature across different scenarios. We compared average and maximum sea surface temperatures across scenarios with the Kruskal–Wallis test, followed by the Wilcoxon paired test to detect specific differences between scenarios.

## RESULTS

We identified 14 bacterial taxa associated with crustaceans and macroalgae (Table 1). The squat lobster, *G. gregaria*, was the host that had the highest number (nine) of cultivable bacterial taxa, followed by the brown macroalgae *M pyrifera* (four) and the centolla *L. santolla* (two).

*Vibrio* bacteria were found in all hosts, whereas both crustaceans shared *Shewanella*. Cultivable epibiotic bacteria in the squat lobster included *Brachybacterium* sp. and *Staphylococcus equorum*. Interestingly, *S. equorum* colony had similar morphological characteristics to *Vibrio* colonies (yellow-orange color and intense growth in TCBS culture medium). Cultivable epibiotic bacteria in the brown macroalgae included *Photobacterium profundum*, *Psychromonas antartica*, and *Cellulophaga* sp. M8-3.

*Vibrio* bacteria reported in this study were phylogenetically close to each other (Fig. 1). The closest relationships with reference sequences were with *V. echinoideorum*, *V. tasmaniensis*, and *V. atlanticus*. Intriguingly, pathogenic bacteria, such as *V. cholerae* and *V. parahaemolyticus* were located in a contiguous clade, while *V. vulnificus* was the most distant pathogenic bacteria.

Statistical comparisons revealed different patterns in mean and maximum surface temperatures in future decades (Fig. 2, Table 2). Mean surface temperature will differ across scenarios (KW = 88.944, $p < 0.001$), but only between current temperatures and those expected between 2050 and 2100 in both scenarios: SSP26 (p.adjusted = 0.042) and SSP45 (p.adjusted = 0.007). Current mean temperatures were on average 7.28 °C, while mean temperatures in the two periods 2020–2050 and 2050–2100 will be similar; 7.56 °C (2020–2050) and 7.81 °C (2050–2100) in the SSP26 scenario, whereas 7.48 °C (2020–2050) and 8.06 °C (2050–2100) in the SSP45 scenario.

**Table 1  Cultivable epibiotic bacteria of marine hosts according to BLAST alignments with the 16S rRNA gene database.**

| Host | Class | Order | Family | Genus | Species | Reference accession number/BLAST identity (%) | NCBI accession number |
|---|---|---|---|---|---|---|---|
| Squat lobster *Grimothea gregaria* | Gammaproteobacteria | Vibrionales | Vibrionacea | *Vibrio* | *Vibrio tasmaniensis strain B1* | KF444394.1/99.78 | PP266351 |
| | | | | | *Vibrio sp. strain B2* | KF444393.1/99.50 | PP266352 |
| | | | | | *V. tasmaniensis strain B8* | KF444400.1/99.78 | PP266358 |
| | | Enterobacterales | Shewanellaceae | *Shewanella* | *S. electrodiphila* | KX078084.1/99.70 | PP266355 |
| | Bacilli | Bacillales | Staphylococcaceae | *Staphylococcus* | *S. equorum strain T1Z32* | OQ472446.1/99.67 | PP266353 |
| | | | | | *S. equorum* | MK015791.1/99.86 | PP266359 |
| | Actinobacteria | Micrococcales | Dermabacteraceae | *Brachybacterium* | *Brachybacterium sp.* | MT163354.1/99.28 | PP266354 |
| | | | | | *B. faecium strainXJB-YJ8* | KM186613.1/99.57 | PP266356 |
| | | | | | *Brachybacterium sp.* | MF092313.1/99.49 MT043879.1/99.35 | PP266357 |
| Centolla (southern king crab) *Lithodes santolla* | Gammaproteobacteria | Vibrionales | Vibrionacea | *Vibrio* | *V. tasmaniensis strain B10* | EU862332.1/99.79 | PP266360 |
| | | Alteromonadales | Shewanellaceae | *Shewanella* | *Shewanella sp. MA325* | FR744883.1/99.71 | PP266361 |
| Brown macroalgae *Macrocystis pyrifera* | Gammaproteobacteria | Vibrionales | Vibrionacea | *Photobacterium* | *P. profundum* | DQ027054.1/99.86 | PP266362 |
| | | | | *Vibrio* | *V. atlanticus strain B14* | LN832929.1/99.51 | PP266363 |
| | | | | | *Vibrio sp. strain B16* | KF444400.1/99.78 | PP266365 |
| | | Alteromonadales | Psychromonadaceae | *Psychromonas* | *Psychromonas antarctica* | AY771768.1/99.28 | PP266364 |

Similarly, maximum surface temperatures in future decades will differ across scenarios (KW = 88.944, $p < 0.001$), but all expected conditions have significant variation among them. Current maximum temperatures were on average 11 °C, while maximum temperatures in the two periods 2020–2050 and 2050–2100 will be; 12.26 °C (2020–2050) and 12.39 °C (2050–2100) in the SSP26 scenario, whereas 12.07 °C (2020-2050) and 12.72 °C (2050–2100) in the SSP45 scenario. Interestingly, maximum surface temperatures above 14 °C are expected to occur only during scenario SSP45 (2050–2100).

## DISCUSSION

We report for the first time some cultivable epibiotic bacteria from centolla, squat lobster, and brown macroalgae from the Magellan Strait, Chile. Our results represent a valuable contribution to the distribution and ecology of *Vibrio* bacteria in marine hosts in a subantarctic ecosystem. In addition, we explored the thermal suitability of the Magellan Strait in future decades under two climate change scenarios. Oceanographic predictions gave insight into the future thermal characteristics of the Magellan Strait. They suggest that: (1) Mean surface temperature will be similar between contemporary values and those projected in the next 30 years (2020–2050), in contrast, it will differ from those in the last 50 years of the century (2050–2100). (2) Maximum sea surface temperature will increase on average by 1.26 °C during 2020–2050 (SSP26), and by 1.39 °C during 2050–2100 (SSP45) in comparison to contemporary sea surface maximum temperatures. (3) The maximum

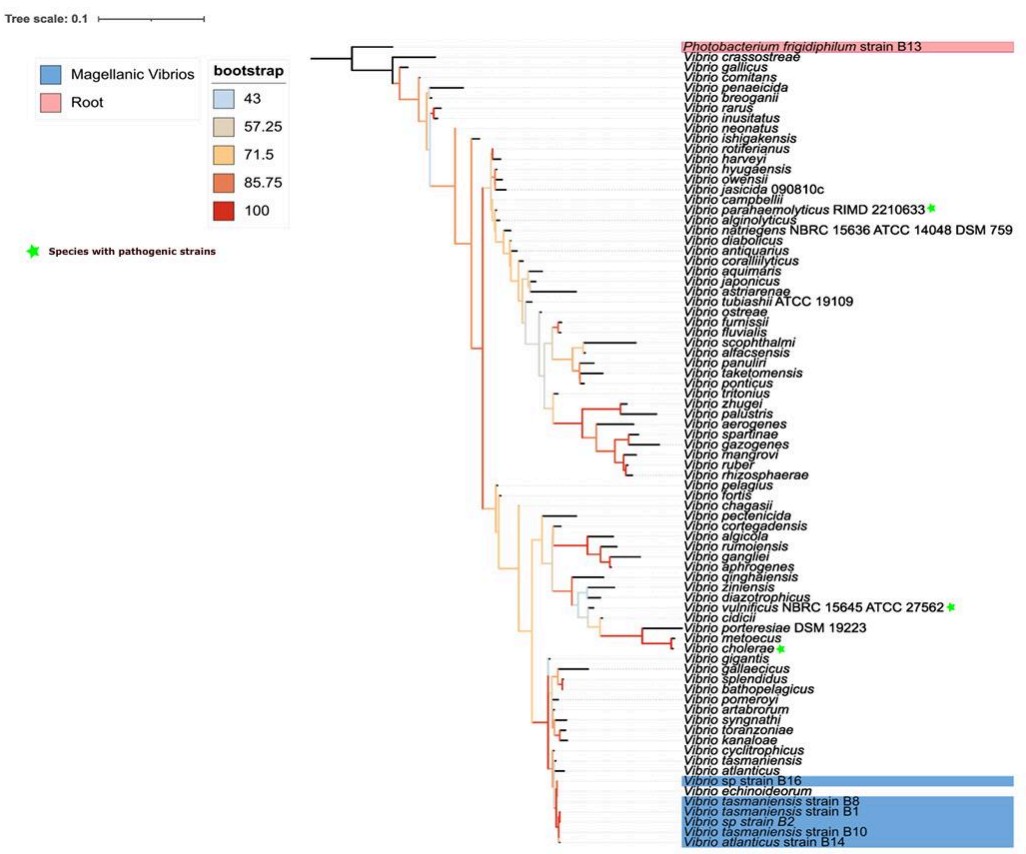

**Figure 1** **Phylogenetic context of epibiotic *Vibrio*.** Phylogenetic tree of 16S rRNA gene from different *Vibrio* species. The tree is unrooted, best-fit model: TIM3e+I+G4 chosen according to Bayesian inferences criterion (BIC). Sequences corresponding to isolates from Magallanes are those with the NCBI GeneBank identifier at the beginning of the branch name.

sea surface temperature record will occur during 2050–2100 in the SSP45, where there could occur sea surface temperatures above 14 °C. These expected temperatures fall within the thermic niche of important pathogenic *Vibrio* species, which could create suitable environmental conditions for their proliferation.

## Latitudinal expansion of the distribution of *Vibrio* bacteria in Chile

*Vibrio* bacteria associated with marine hosts in Chile were reported in fish in Quintay Bay (33°11′S; 71°1′W) in Valparaíso (*Lasa et al., 2015*) and mollusks in Tongoy Bay (30°15′S 71°29′W) in Coquimbo (*Rojas et al., 2019*). Recent reports have indicated the presence of *Vibrio* bacteria in some hosts, such as the nest soil and body surfaces of Magellanic penguins (*Ochoa-Sánchez et al., 2023b*; *Ochoa-Sánchez et al., 2024*), and as a pathogen of farmed Atlantic salmon (*Bohle et al., 2007*). Also, *Vibrio* bacteria are common microbes of the marine microbial community in the Magellan region (*Crisafi et al., 2010*; *Maturana-Martínez et al., 2021*). Hence, it is likely that there is a marine reservoir of autochthonous subantarctic *Vibrio* bacteria that could interact with marine hosts. However, our results

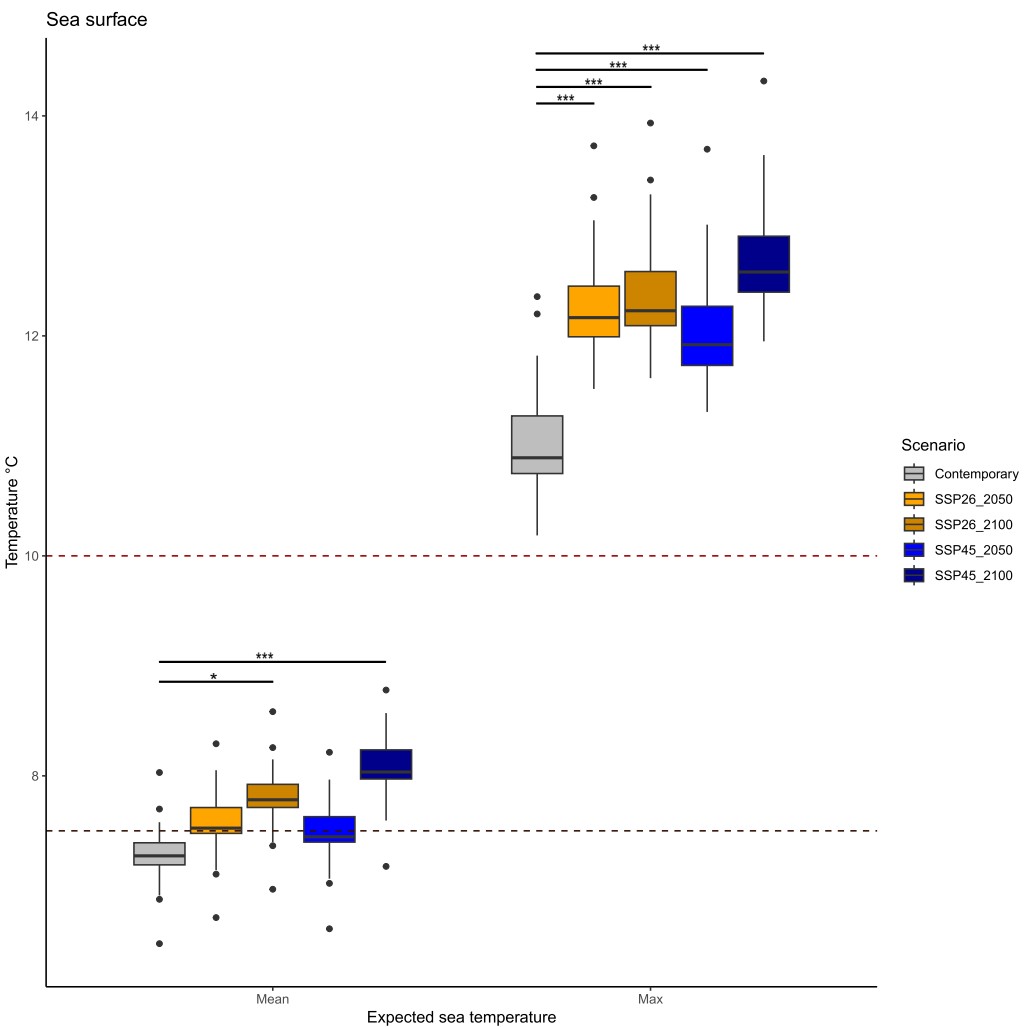

**Figure 2** **Expected average and maximum sea surface temperature in the Magellan Strait across different scenarios, including contemporary measures and values from four climate change scenarios (SSP26_2050, SSP26_2100, SSP45_2050, and SSP45_2100).** Boxplots depict the first (Q1), second (median), and third (Q3) quartile, while whiskers depict the interquartile range (Q1−1.5) and (Q3 + 1.5). Statistical notation, $* = p < 0.05$, $** = p < 0.001$, $*** = p < 0.0001$. Black points are values that fall outside the interquartile range. Dashed lines represent the average growth temperature of two pathogenic *Vibrio* species from cold environments: black line 7.5 °C (*Vibrio tasmaniensis*) and red line 10 °C (*Vibrio splendidus*).

indicate the presence of *Vibrio* bacteria in the epibiotic microbiota of three marine wild hosts in the Magellan Strait: the squat lobster, the centolla, and the brown macroalgae.

## On the potential role of increasing temperatures in epibiotic subantarctic *Vibrio*

Over the last 30 years, *Vibrio* infections have been positively correlated with sea surface temperature increases (*Jayakumar et al., 2024*). Also, rising sea surface temperatures are associated with increased *Vibrio* pathogenicity. For example, increases of 4 °C in sea

**Table 2  Mean and range of average and maximum sea surface temperatures across different climate change scenarios.**

| Climatic scenario | Average sea surface temperatures | | Maximum sea surface temperatures | |
|---|---|---|---|---|
| | Mean | Range | Mean | Range |
| SSP26 2020–2050 | 7.56 | 6.71–8.29 | 12.26 | 11.51–13.72 |
| SSP26 2050–2100 | 7.81 | 6.96–8.58 | 12.39 | 11.61–13.93 |
| SSP45 2020–2050 | 7.48 | 6.60–8.21 | 12.07 | 11.30–13.69 |
| SSP45 2020–2100 | 8.06 | 7.17–8.78 | 12.72 | 11.95–14.31 |

surface temperature increase clam mortality by *Vibrio* by 70% (*Tian et al., 2024*). Also, marine heatwaves are associated with increased *Vibrio* pathogenicity in oysters (*Siboni et al., 2024*). Under ongoing climate change, it is important to have scientific data of expected increases in sea-surface temperature, which could inform about the pathogenic risk for aquaculture activities from marine microbes. Our results indicate that the expected average and maximum average sea surface temperatures in the Magellan Strait would be around 7–8 °C and around 12.26–12.72 °C, respectively. Hence, given the presence of native *Vibrio* in the waters of the Magellan region (*Crisafi et al., 2010*; *Maturana-Martínez et al., 2021*) and the role of rising sea surface temperatures in *Vibrio* metabolism, prevalence, and pathogenicity, it is plausible that *Vibrio* bacteria from subantarctic ecosystems will have higher metabolic activity in the future.

However, there are cases when *Vibrio* bacteria could produce disease under cold conditions. For example, *Vibrio splendidus*, a sister species to *Vibrio atlanticus* (found on the surface of macroalgae), has caused disease in the Yesso scallop (*Patinopecten yessoensis*) in northeastern China at 10 °C (*Liu et al., 2013*). Similarly, another strain of *Vibrio splendidus* has caused disease in the gorgorian octocoral *Eunicella verrucosa*, in England, at 7.5 °C (*Vattakaven et al., 2006*). Furthermore, important pathogenic *Vibrio* species, such as *V. parahaemolyticus* and *V. vulnificus*, could grow even at refrigeration with temperatures below 10 °C (*Kim et al., 2012*). Noteworthy, *V. parahaemolyticus* could grow inside oysters below 15 °C (*Fernandez-Piquer et al., 2011*). *Vibrio parahaemolyticus* might be particularly relevant for the subantarctic Chilean region since epidemic outbreaks of this species were recorded a decade ago in Puerto Montt, Chile (*Raszl et al., 2016*), approximately 10° latitudinal degrees above the Magellan Strait. The fact that some *Vibrio* species could produce disease under cold conditions, coupled with the expected increases in sea surface temperature in the Magellan region, as well as the intensive aquaculture activities, encourages the establishment of a constant microbiological program to prevent epidemiological outbreaks.

### *Vibrio tasmaniensis* as a common member of the epibiotic bacteria in centolla and squat lobsters

We detected different strains of *Vibrio tasmaniensis* on the surface of crustaceans, strains B1 and B8 in the squat lobster, and strain B10 in the centolla. *Vibrio tasmaniensis* is commonly found as a commensal microbe of mollusks (*Travers et al., 2015*), however,

there are pathogenic strains within this species. For example, *V. tasmaniensis LGP32* is an intracellular pathogenic strain of the oyster *Crassostrea gigas*, that infects its hemocytes and causes hemocyte cytotoxicity (*Vanhove et al., 2016*). Moreover, infection caused by *V. tasmaniensis LGP32* leads to systemic infection that damages connective tissue, muscles, and several organs (*Rubio et al., 2019*). The strains of *V. tasmaniensis* inhabiting the squat lobster and the centolla are likely innocuous commensals. However, innocuous commensals can turn into pathogens upon acquisition of plasmids with virulence factors, as has been shown in the commensal *Vibrio crassostreae* associated with oysters, a phylogenetically closely related bacterium to *V. tasmaniensis* (*Bruto et al., 2017*). Conversion of innocuous strains into pathogenic strains has also been described in *Vibrio vulnificus* (*López-Pérez et al., 2021*). In this case, pathogen emergence is caused by horizontal transmission of the capsular polysaccharide cluster, an important virulence factor in *V. vulnificus* (*Pettis & MukerjiA, 2020*).

## Host ecology relevance in *Vibrio* prevalence and future ecological perspectives

The life-history strategy of centolla naturally increases their probability of encountering and spreading *Vibrio*. Centolla usually eats algae or crustaceans (cannibalism is common) (*Andrade et al., 2022*), hosts where *Vibrio* commonly grows as biofilm (*Vezzulli et al., 2010*). From the spatial ecology perspective, juvenile centollas usually gather in high densities upon brown macroalgae rhizoids (*Cárdenas et al., 2007*), which could facilitate dispersion of *Vibrio* among juvenile king crabs. From the phenological and reproductive ecology perspective, adult king crabs court during spring in shallow waters (>20 m depth) (*Lovrich et al., 2002*). These behavioral and phenological factors might expose centolla to the highest SST max. Besides, ongoing sea surface warming is expected to impair the physiology of king crabs. Temperatures above 12 °C produce physiological stress in larvae and juveniles of centollas (*Calcagno et al., 2005*). Furthermore, *Vibrio* bacteria are the main bacterial pathogens of crustaceans (*De Souza Valente & Wan, 2021*), and the activity of pathogenic species is highly correlated with seasonal peaks of sea surface temperature (*Vezzulli, Colwell & Pruzzo, 2013*).

The expected increase in the SST max of the region may have important consequences for fisheries and aquaculture practices. Temperature is a multifactorial driver of *Vibrio* ecology. Although tropical *Vibrio* strains require SST max above 18 °C to proliferate, important pathogenic *Vibrio* could start growing at 10 °C (*McGovern & Oliver, 1995*; *Kirschner et al., 2008*; *Kim et al., 2012*). Furthermore, horizontal transmission of virulence factors underlies the fast evolution of *Vibrio* bacteria from innocuous to pathogenic strains in a few years (*López-Pérez et al., 2021*). Alternatively, natural weather events could facilitate the dispersion of pathogenic clades across oceans, as occurred with an Asiatic lineage of *V. parahaemolyticus* that migrated to South America during El Niño event in 1997, due to the arrival and displacement of warm water (*González-Escalona et al., 2015*). Notably, *V. parahaemoliticus* appears to be the most successful *Vibrio* spp. in the southern region of South America, since it has caused several epidemic outbreaks of hemorrhagic diarrhea in the last 20 years in Puerto Montt, Chile (located at 42°S) (*Raszl et al., 2016*).

The reported *Vibrio* bacteria in this study lie in a contiguous clade to the phylogenetic clade where lie two of the most important *Vibrio* species from the clinical perspective: *V. cholerae* and *V. vulnifius*. Also, we detected *V. tasmaniensis* in crustaceans, a *Vibrio* species that has known pathogenic strains in invertebrates (*Vanhove et al., 2016*; *Rubio et al., 2019*). This result *per se* does not inform about the pathogenic potential of the *Vibrio* bacteria found in this study, however, phylogenetic relatedness could increase the probability of successful genetic exchange that could lead to pathogen emergence, which could be a likely scenario in future decades due to increasing sea surface temperatures and the link between high temperatures and genetic horizontal transmission (*Kim et al., 2012*; *López-Pérez et al., 2021*). Hence, we strongly suggest starting a comprehensive, continuous monitoring program in the Magellan Region centered on *Vibrio* detection and its genomic repertoire, ideally upon several potential hosts.

## Occurrence of non-*Vibrio* bacteria in crustaceans

The non-*Vibrio* bacteria we cultivated from the surface of crustaceans were *Brachybacterium* sp. (Dermabacteraceae: Actinomycetota), *Shewanella* sp. SS15 (Shewanellaceae: Proteobacteria), and *Staphylococcus equorum* (Staphylococcaceae: Bacillota). These bacteria are common members of marine and cold environments across the world (*Lemaire, Mejean & Iobbi-Nivol, 2020*; *Ribeiro et al., 2023*). *Shewanella sp.* bacteria are common epibiotic bacteria of marine hosts, whose role might be host-specific. For example, several epibiotic *Shewanella* taxa are capable of synthesizing tetrodotoxin, a potent neurotoxin, which prevents predation from its hosts (*Magarlamov, Melnikova & Chernyshev, 2017*). Furthermore, other *Shewanella* bacteria can produce a wide array of metabolites that interfere with *Vibrio* spp. proliferation (*Zhang et al., 2023*). Hence, it could have a beneficial role against the proliferation of pathogenic *Vibrio*. We want to point out that *S. equorum* (isolated from *G. gregaria*) had a colony phenotype strongly resembling the *Vibrio* colonies (orange/red colonies), underscoring the importance of incorporating molecular approaches to identify culturable bacteria.

## Limitations of the study

Our results are biased by the sampling method, which focused on the use of a neutral marker gene from culturable microorganisms. Furthermore, the low sample size of our study and regional tissue microbiological variation limit the generalizability of the presence of *Vibrio* bacteria in these species. Thus, it is uncertain the quantitative, metabolic, and pathogenic relevance of *Vibrio* bacteria in crustaceans and macroalgae from the Magellan Strait. However, we believe this is a valuable contribution to the distribution and ecology of *Vibrio* bacteria in subantarctic environments. Further studies should combine higher sample sizes with genomic approaches to test the prevalence of *Vibrio* bacteria in these hosts and to characterize their metabolic repertoire, which should properly inform about their ecological relevance and pathogenic potential.

## CONCLUSIONS AND PERSPECTIVES

These results highlight the presence of native epibiotic *Vibrio* from crustaceans and macroalgae in a subantarctic ecosystem, the Magellan Strait, which increases the natural distribution of this bacterium commonly found in tropical and temperate waters. Interestingly, we found several strains of *V. tasmaniensis* in the centolla and the squat lobsters, which increases the geographic distribution and host ecology of this bacterium. *Vibrio* bacteria reported in this study lie in a contiguous clade to some pathogenic *Vibrio*, such as *V. cholerae* and *V. vulnificus*; however, further genomic approaches are required to better inform about the pathogenic potential of these strains.

Oceanographic predictions estimate that future average values of mean and maximum sea surface temperature will be in the range of 7.56–8.06 and 12.26–12.72, depending on the climatic scenario and period. However, maximum sea surface temperature could have specific points where temperatures rise above 14 °C in the SSP45 during 2050–2100. This expected seasonal increase in sea surface temperatures, coupled with the high abundance of crustacean hosts, could increase the activity of native *Vibrio* from the Magellan Strait, which could have relevant consequences for aquaculture activities. In consequence, we strongly suggest starting a comprehensive, continuous monitoring program in the Magellan Region centered on *Vibrio* detection and its genomic repertoire, ideally upon several potential hosts. Furthermore, given the possibility to cultivate *Vibrio*, it becomes possible to conduct experimental studies that address the effect of expected warming and UV radiation in psychrophilic *Vibrio*, as well as their pathogenic and antibiotic potential. Finally, *Vibrio* monitoring in subantarctic ecosystems could serve as a microbial biosensor that attests the progress of climate change (*Baker-Austin et al., 2017*).

## ACKNOWLEDGEMENTS

We are grateful to Helena S. Hernández-Rosas for technical assistance. We thank captains Hugo Cárdenas and Ademir Añazco and their crews of the ship "MaryPaz II" and "Arturo 2", respectively, for their safe travels and assistance during sampling.

### Funding

This work was funded by Proyecto de Continuidad CEQUA and ANID project number R20F0009. The funders had no role in study design, data collection and analysis, decision to publish, or preparation of the manuscript.

### Grant Disclosures

The following grant information was disclosed by the authors:
Proyecto de Continuidad CEQUA and ANID project number R20F0009.

### Competing Interests

Luis E. Eguiarte and Valeria Souza are Academic Editors for PeerJ.

## Author Contributions

- Manuel Ochoa-Sánchez analyzed the data, prepared figures and/or tables, authored or reviewed drafts of the article, and approved the final draft.
- Rosalinda Tapia-López performed the experiments, analyzed the data, prepared figures and/or tables, authored or reviewed drafts of the article, and approved the final draft.
- Ulises E. Rodriguez-Cruz analyzed the data, prepared figures and/or tables, authored or reviewed drafts of the article, and approved the final draft.
- Eliana Paola Acuña Gomez conceived and designed the experiments, authored or reviewed drafts of the article, collection of biological material, and approved the final draft.
- Luis E. Eguiarte conceived and designed the experiments, authored or reviewed drafts of the article, collection of biological material, and approved the final draft.
- Valeria Souza conceived and designed the experiments, authored or reviewed drafts of the article, collection of biological material, and approved the final draft.

## Field Study Permissions

The following information was supplied relating to field study approvals (*i.e.*, approving body and any reference numbers):

Samples were collected following approved bioethical guidelines of the Comité de Ética, Bioética y Bioseguridad, Universidad de Concepción, Chile (protocol number CEBB 1081 2021) and a sampling permit of the Subsecretaría de Pesca y Acuicultura, Chile (resolution E- 2021- 531).

## DNA Deposition

The following information was supplied regarding the deposition of DNA sequences:

Raw sequences are available at GenBank: PP266351 to PP266365.

## Data Availability

The raw 16S sequences are available at GenBank: PP266351.1, PP266352, PP266358, PP266355, PP266353, PP266359, PP266354, PP266356, PP266357, PP266360, PP266361, PP266362, PP266363, PP266365, PP266364.

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
