# Peer review of "Epibiotic Vibrio bacteria from crustaceans and macroalgae in a subantarctic ecosystem, and their future thermal suitability"

_PeerJ, doi:10.7717/peerj.19881_

## Round 0.1 · original submission · Major Revisions

· Academic Editor

Major Revisions

Please find the enclosed reports by the reviewers. When resubmitting your manuscript for consideration, please respond to all the reviewers' comments and concerns in a point-by-point manner.

Reviewer 1 ·

Basic reporting

Ochoa-Sánchez et al. cultured 14 different bacterial taxa from a crab, a squat lobster and kelp from waters of the Strait of Magellan, Chile. The manuscript combines a culturing assessment of the three previously mentioned host species with a predictive study of sea surface temperature in the region and the authors speculate what the potential impact will be on Vibrio spp. in the region. I found the manuscript disconnected and illogical. No empirical information is provided on the impact of sea surface temperature on Vibrio spp. The authors, for example, could have used an experimental approach using microcosms to simulate future climate scenarios and the impact on Vibrio assemblages. All the authors did was to culture several bacterial taxa, including some classified as belonging to the genus Vibrio, from three host species. No additional information is, however, provided about these strains. It would have been more interesting to test strain growth against different culturing conditions including the preferred temperature tolerances in terms of minimum and maximum temperature thresholds. No hypotheses were presented or tested during this study and although the study itself is purely descriptive in nature, the authors did a very poor job of properly describing the results as pertains to the cultured strains. In my opinion, this manuscript should not have been sent for review, but should have been rejected at the editorial stage.

Minor comments

Lines 90-91: two times South America
Line 133: channel prawns?
Line 264: Crassoktrea

Experimental design

No comment

Validity of the findings

No comment

Additional comments

No comment

Reviewer 2 ·

Basic reporting

Title
The title emphasizes "thermal suitability" and "subantarctic ecosystems," but the introduction does not sufficiently connect these elements. The authors should clearly link the ecological context of the Strait of Magellan to the potential impacts of future thermal changes on epibiotic bacteria.

Introduction
• Several paragraphs are overly general, and many statements are unsupported by examples.
• Include more recent references when discussing the potential effects of climate change on the distribution of this group of bacteria. It is essential to indicate the current state of knowledge in this area.
• There are frequent inconsistencies in terminology, both in the abstract and introduction. For example:
- The study area is alternately referred to as the Strait of Magellan and the Magellan region (I have clarified these differences in one of my comments).
- Similarly, there is inconsistency when referring to the species studied and temperature-related findings (see comments in the manuscript for more detail).

Materials and Methods (M&M)
Sub-section: Marine host sampling, epibiotic bacterial isolation, and DNA extraction
• This section should be rewritten for better clarity and structure.
• The order of ideas is unclear, making it difficult to follow the methodology.
Key details are missing, such as:
- When sampling was conducted (time of year and year of collection).
- When the cultures were transferred to the laboratory for DNA extraction.
• Organizing the section chronologically and providing these missing details would significantly improve its readability.

Sub-section: Sea surface temperature projections under climate change scenarios
• This section should also be rewritten for clarity.
• There are inconsistencies regarding the study area. The manuscript alternates between the Strait of Magellan and the Magellan region. Additionally, the temperature data for each climate scenario appear to correspond to a different area. The study area needs to be clearly defined and consistently referred to throughout the text.
• The analysis involving three species must be explicitly tied to the study area to avoid confusion.
Results
Table 2: The title for Table 2 is missing. Please ensure all tables have descriptive titles.
Figure 2: Since the temperature results are derived from a well-known database (Bio-Oracle) and do not represent original data from your study, I recommend making better use of these results.
For example, add lines to the figure that indicate the known thermal limits of some of the Vibrio species identified in your study. This would provide readers with valuable information about how future temperature changes could create favorable thermal conditions for the presence and proliferation of these bacteria.
Discussion
I was unable to continue my review beyond this point due to the significant revisions needed in the earlier sections (Introduction, Materials and Methods, and Results). These sections require substantial editing before moving forward with the rest of the manuscript. Please refer to the detailed comments provided in the manuscript.
In conclusion, while the manuscript addresses an important topic, it requires significant revisions to improve clarity, consistency, and the presentation of results. I encourage the authors to address these issues before resubmission.

Experimental design

Comments on the manuscript

Validity of the findings

The findings presented in the manuscript have potential scientific value, particularly in addressing a knowledge gap related to Vibrio bacteria in subantarctic ecosystems. However, the validity of the conclusions is undermined by several issues that need to be addressed:

Use of Outdated Climate Data:
The reliance on older versions of Bio-Oracle data (Version 2) and outdated climate scenarios (e.g., RCPs instead of SSPs) limits the applicability of the findings to current and future environmental conditions. Updating the analysis with Version 3 and SSP scenarios would strengthen the validity of the projections.

Incomplete Discussion of Findings:
The manuscript does not adequately explore the ecological implications of the results. For example, the potential impact of thermal changes on Vibrio proliferation could be better supported by integrating known thermal limits for these bacteria.

General Data Interpretation:
The decision to use maximum surface temperature instead of average values, without adequate justification, could lead to biased or incomplete interpretations. Clarifying this choice and its impact on the results would improve the credibility of the findings.

Additional comments

Recommendations
The manuscript has potential but requires substantial revisions before it can be considered for publication. The authors should address the following critical issues:

Reorganize sections for better clarity and logical flow.
Strengthen scientific claims by incorporating updated datasets, methods, and references.
Clarify and standardize terminology throughout the manuscript.
Provide detailed explanations for methodological choices, particularly regarding temperature metrics and climate scenarios.

Final Decision
Major revisions required. The manuscript will be suitable for publication only after significant reworking to improve clarity, consistency, and scientific rigor.

Annotated reviews are not available for download in order to protect the identity of reviewers who chose to remain anonymous.

Reviewer 3 ·

Basic reporting

The manuscript is written in clear and professional English.
-The introduction provides sufficient context and establishes the relevance of studying native Vibrio bacteria in marine hosts within a subantarctic ecosystem
-The manuscript cites relevant and current literature; however, the bibliography must be reorganized in alphabetical order and the referencing style should be standardized.
-All figures are relevant to the study and well-labeled, but the resolution and readability should be improved for better comprehension.

Experimental design

This is an original study within the scope of the journal, exploring the role of Vibrio bacteria in a subantarctic ecosystem
-The methods are generally well-documented, enabling replication by other researchers.
-The study adheres to ethical standards appropriate for microbiological and environmental research

Validity of the findings

The data is robust and statistically sound. The manuscript includes appropriate tests of significance) to support its findings
- The conclusions are well-supported by the data and are clearly linked to the research question
- The study’s design and presentation encourage replication, particularly due to its detailed methods and data availability

Additional comments

The authors could provide more details about the handling of the swabs after collecting the samples from the crustaceans and macroalgae, prior to their inoculation onto TCBS agar.

-In some cases, TCBS agar inhibits bacterial growth due to its composition. Did you have any problems using only TCBS agar? Particularly in environmental studies may occur problems with selectivity with this medium.

-The typography in figure 1 is unreadable at that scale. Would it be possible to make it in a larger scale?

-Refences
Numbers 7, 10, 27 and 37 are not found in the text.


In particular,

Line 144, 149, 226, 228, 232, 296, 297, 343
Check if the temperature is indicated as XX°C, XX °C or XX ° C

Line 180-181
Minh et al., 2020 include it in the references

Line 255
Maturana-Martínez et al. 2020. Check because in the bibliography the year is 2021

Lines 168-169
"The 16 sequences obtained were then submitted to GenBank with the following accession numbers PP266351 to PP266365". In table 2 there are only 15 records

---

## Round 0.2 · Major Revisions

· Academic Editor

Major Revisions

Please revise your manuscript by following the reviewers' comments. Please make sure that a point-by-point response letter is provided when submitting your revised manuscript.

Reviewer 4 ·

Basic reporting

As far as I understand, the study's objective was to identify epibiotic bacteria of marine organisms in the Strait of Magellan area, a subantarctic ecosystem, and to predict the presence or increase of potentially pathogenic bacteria due to rising water temperatures. However, there seem to be some gaps that need to be addressed. Could the discovery of Vibrio bacteria now indicate a change in the epibiont microbiota? How can you know if there's no history of their presence before? Are these bacteria exclusive to these organisms, or are they also found in water? What impact does their presence have on these organisms in particular?

I agree with Reviewer 1's comments on the previous review. Although the authors made several corrections suggested by the other reviewers, I consider the manuscript neither innovative nor an appropriate contribution to this journal. I do not understand how they relate sea surface temperature predictions to the presence of Vibrio spp. Perhaps if they had a history of the abundance of this genus when temperatures were colder, it would make more sense. There is evidence of Vibrio recorded in seawater in this region since 1995 that could support their hypothesis, although I consider it to be very weak.
It would also be interesting to consider the results of Crisafi et al (2010), who reported the presence of Vibrio in water samples from the Strait of Magellan since 1995. They mention that the abundance was low, which would also be interesting to discuss according to your hypothesis that the change in water temperature could increase the presence of these bacteria.

Crisafi, E., Azzaro, M., Lo Giudice, A. et al. Microbiological characterization of a semi-enclosed sub-Antarctic environment: the Straits of Magellan. Polar Biol 33, 1485–1504 (2010). https://doi.org/10.1007/s00300-010-0836-6

Experimental design

The isolation and identification of bacteria follow the appropriate methodology and are clearly explained.
The temperature prediction seems disconnected to me; I can't understand how this relates to the isolated bacteria results. Perhaps more detailed information would help to understand.

Validity of the findings

The manuscript has potential, but requires substantial revisions before it can be considered for publication. The authors should provide more detailed explanations, particularly regarding the relationship between temperature and climate scenarios and the presence of Vibrio; these results need to be further connected.

Reviewer 5 ·

Basic reporting

This interesting study examines the microbiome associated with two species of crabs, Lithodes santolla, and Grimothea gregaria, and the bacteria communities of brown macroalgae Macrocystis pyrifera, emphasizing how climate change could be increasing the risk of potentially pathogenic bacteria, underscoring the importance of understanding and documenting local bacterial biodiversity, although
the purpose of the study is not entirely clear to me. ¿Was the main objective to characterize the bacterial communities associated with three different basibionts?

Experimental design

The rationale behind the small sample size (4 individuals per crustacean species and 2 Macrocystis blades) was not clearly explained. It would be helpful to provide justification for this limited sampling or to acknowledge its implications for the validity of the findings

Validity of the findings

The study provides valuable preliminary insights into the composition of epibiotic bacterial communities
Future research with larger sample sizes will be essential to confirm these trends and explore their ecological relevance

Additional comments

The presentation of this study feels somewhat scattered and would benefit from a clearer structure. The main objectives of the work are not entirely evident, beyond reporting the presence of bacteria associated with marine organisms for the first time in the Magellan Strait. Clarifying the research questions and goals early in the manuscript would help guide the reader and strengthen the overall contribution.

Annotated reviews are not available for download in order to protect the identity of reviewers who chose to remain anonymous.

---

## Round 0.3 · accepted · Accept

· Academic Editor

Accept

The manuscript can be accepted for publication now.

Reviewer 4 ·

Basic reporting

The authors made substantial changes to the manuscript to clarify their objectives and findings, so that they now also clearly address their hypothesis. The manuscript is well written, and the bibliographic references are sufficient and appropriate. The document's structure is correct and professional, as are the graphs and tables.

Experimental design

The methods are described in detail so that they can be replicated.

Validity of the findings

The results of this research are well described, thoroughly discussed, and presented with a clear conclusion.

Additional comments

I thank the authors for their responses to the questions I posed in the previous review and for considering the suggestions I made regarding their manuscript.